# Comment on 'AIRE-deficient patients harbor unique high-affinity disease-ameliorating autoantibodies'

Nils Landegren[1,2]*, Lindsey B Rosen[3], Eva Freyhult[4,5], Daniel Eriksson[1,6], Tove Fall[7], Gustav Smith[8,9,10], Elise M N Ferre[3], Petter Brodin[8,11,12], Donald Sharon[13], Michael Snyder[10,13], Michail Lionakis[3], Mark Anderson[14], Olle Kämpe[1,6,15]

[1]Department of Medicine (Solna), Karolinska University Hospital, Karolinska Institutet, Stockholm, Sweden; [2]Department of Medical Sciences, Science for Life Laboratory, Uppsala University, Uppsala, Sweden; [3]Laboratory of Clinical Immunology & Microbiology, National Institute of Allergy and Infectious Diseases, National Institutes of Health, Bethesda, United States; [4]Department of Medical Sciences, National Bioinformatics Infrastructure, Uppsala, Sweden; [5]Science for Life Laboratory, Uppsala University, Uppsala, Sweden; [6]Department of Endocrinology, Metabolism and Diabetes, Karolinska University Hospital, Stockholm, Sweden; [7]Department of Medical Sciences, Molecular Epidemiology, Science for Life Laboratory, Uppsala University, Uppsala, Sweden; [8]Department of Cardiology, Clinical Sciences, Lund University, Skåne University Hospital, Lund, Sweden; [9]Program in Medical and Population Genetics, Broad Institute of Harvard, Massachusetts Institute of Technology, Cambridge, United States; [10]Wallenberg Center for Molecular Medicine, Lund University Diabetes Center, Lund University, Lund, Sweden; [11]Department of Women's and Children's Health, Science for Life Laboratory, Karolinska Institutet, Stockholm, Sweden; [12]Department of Newborn Medicine, Karolinska University Hospital, Stockholm, Sweden; [13]Department of Genetics, School of Medicine, Stanford University, Stanford, United States; [14]Diabetes Center, University of California, San Francisco, San Francisco, United States; [15]KG Jebsen Center for Autoimmune Diseases, University of Bergen, Bergen, Norway

*For correspondence:
nils.landegren@ki.se

**Abstract** The *AIRE* gene plays a key role in the development of central immune tolerance by promoting thymic presentation of tissue-specific molecules. Patients with *AIRE*-deficiency develop multiple autoimmune manifestations and display autoantibodies against the affected tissues. In 2016 it was reported that: i) the spectrum of autoantibodies in patients with *AIRE*-deficiency is much broader than previously appreciated; ii) neutralizing autoantibodies to type I interferons (IFNs) could provide protection against type 1 diabetes in these patients (Meyer et al., 2016). We attempted to replicate these new findings using a similar experimental approach in an independent patient cohort, and found no evidence for either conclusion.
DOI: https://doi.org/10.7554/eLife.43578.001

## Introduction

The immune system must be capable of mounting responses against foreign antigens of all possible sorts while at the same time remaining inert against self-components. Studies of a rare monogenic disorder – autoimmune polyendocrine syndrome type 1 (APS1) or autoimmune polyendocrinopathy-candidiasis-ectodermal dystrophy (APECED) and the mutated gene underlying the disease, Autoimmune regulator (*AIRE*) – have shed light on how reaction against self is avoided. The *AIRE* gene encodes a transcription factor that promotes thymic presentation of genes with limited peripheral tissue distribution (*Anderson et al., 2002*). *AIRE*-mediated antigen exposes naïve T-cells to self-molecules that otherwise would not be present in the thymus, and is therefore critical for the negative selection of autoreactive T-cells. Patients with APS1 develop multiple tissue-destructive manifestations and harbor autoantibodies targeting the affected tissues (*Ahonen et al., 1990*; *Ferre et al., 2016*; *Söderbergh et al., 2004*). A number of *bona fide* autoantigens have been identified in APS1 over the years, many of which are now used routinely in the clinical assessment of patients with APS1 and also in other autoimmune diseases (*Alimohammadi et al., 2008*; *Alimohammadi et al., 2009*; *Ekwall et al., 1998*; *Kisand et al., 2010*; *Landegren et al., 2016a*; *Landegren et al., 2015*; *Meager et al., 2006*; *Puel et al., 2010*; *Shum et al., 2013*; *Winqvist et al., 1993*; *Winqvist et al., 1992*). To gain a more comprehensive view on the autoantigen spectrum in APS1 we used a panel of 9000 full-length human proteins to perform a broad-scale mapping of autoantibody targets (*Landegren et al., 2016b*; *Landegren et al., 2015*). In the proteome array screen we replicated previously reported autoantigens and further identified and validated three novel major autoantigens. We limited our analyses to autoantibody targets that were shared between multiple patients and thereby showed statistically robust association with the patient group. We did not address autoantibody signals against targets that were observed only in one or a few patients, as we could not confidently separate these types of signals from the background of normal variation and technical noise. Indeed, elevated signals of this kind were observed both among patients and among the healthy controls.

Meyer et al. recently reported results from a screening experiment using a similar approach in APS1 patients, where the same protein panel was used to study autoantibodies in a comparable cohort of patients with APS1 and healthy controls (*Meyer et al., 2016*). We were surprised that their description of the autoantigen spectrum was very different from ours. Meyer et al. identified over 3700 proteins as autoantigens in APS1, corresponding to around 40% of the investigated panel. The healthy controls were found to largely lack high-level serum reactivity, which was used as evidence for the relevance of the autoantibody targets identified in the APS1 patients.

As the methods reported in the experiments by Meyer et al. were very similar to those we used, we asked whether differences in the approach to data analysis could explain the different conclusions. We noted that Meyer et al. applied a cutoff definition that could be expected to result in a skewed detection of autoantibody signals between patients and controls. To address this question, we used proteome array data for 51 APS1 patients and 21 healthy controls (*Landegren et al., 2016b*; *Landegren et al., 2015*) to reevaluate the autoantigen spectrum in APS1. In permuted datasets we found that the approach to data analysis used by Meyer et al. was strongly biased towards overestimating the number of autoantibody signals in the patient group. When we applied alternative statistical analyses without this inherent skewing effect, we found no support for the conclusion of broad antigen spectrum in APS1. *Meyer et al. (2016)* also proposed that subjects with APS1 that harbor neutralizing autoantibodies against type I interferons are protected against developing type 1 diabetes, one of the autoimmune manifestations of the disorder. We attempted to validate this finding in a larger cohort of APS1 subjects with type 1 diabetes and could find no evidence to support this proposed association.

## Results

### No evidence for widespread autoantibody reactivity in APS1 patients

We used our previously collected proteome array dataset to reevaluate the autoantigen spectrum in APS1. The data were generated by screening a panel of 9000 full-length human proteins (ProtoArray) with sera from 51 patients with APS1 and 21 healthy blood donors (*Landegren et al., 2016b*; *Landegren et al., 2015*). We followed the analytical approach used as closely as possible from the information available in the published article (*Meyer et al., 2016*). The fluorescence signal intensities were log-transformed and normalized using robust linear modeling based on the human-IgG and anti-human-IgG controls. Control spots and suspected printing contamination signals were sorted out (for further details, see Materials and methods). After filtering, 6540 unique proteins remained.

As a first step to ascertain if our dataset was comparable to the dataset in Meyer et al., we performed the same approach to statistical analysis as described to identify autoantibody signals in the APS1 patients and the healthy controls. Z-scores were calculated for all individuals over all proteins as the number of standard deviations above the mean value for the healthy controls. At low Z-score cutoffs (1, 2 and 3) the number of autoantibody signals was similar for patients and controls. However, at higher cutoffs the patient group deviated from the control group, and at cutoff level $Z \geq 5$, autoantibody signals were only observed among the APS1 patients (*Figure 1b*). Our dataset thereby showed a similar result as the one by Meyer et al. when we used the same statistical analysis, specifically identifying high-level autoantibody signals among the APS1 patients. Investigation of established autoantigens in APS1 provided further support that the datasets in our study and the one by Meyer et al. were comparable (*Figure 1—figure supplement 1*).

As a next step, we asked whether the difference in number of autoantibody signals between patients and controls was true or could be explained by a skewed data analysis. Meyer et al. used a cutoff based on the mean and standard deviation of the values obtained for the healthy controls to identify autoantibody signals. Consequently, elevated signals that occur among the controls increase the cutoff while elevated signals among the patients do not affect the cutoff. It could therefore be expected that elevated signals occurring sporadically in both patients and controls were detected unequally between the groups. To determine if this was the case, we first performed a simulation where we introduced an elevated outlier value in either the APS1 patient group or the control group for one of the proteins in the array. For this example, we chose the protein serum albumin, which did not show elevated signal for any patient or control. When we replaced one of the patient values with a signal value at saturation level (65,000 signal units), the cutoff was not affected and the elevated outlier scored as autoantibody positive at Z-score cutoffs of 3, 4 and 5 as expected (*Figure 1—figure supplement 2*). However, when we introduced an outlier at the same signal level in the control group the cutoff drastically increased, which gave the absurd result that the outlier scored as autoantibody negative at Z = 5. The example thereby revealed that even extremely elevated values in the control group could be missed in the analysis.

We also performed simulations where outliers were introduced in random normal data representing 51 'cases' and 21 'controls', which showed how outliers introduced in the controls were less likely to be detected (*Figure 1—figure supplement 3*). We next used our proteome array dataset to study groups of patients and controls randomly assigned to represent 'cases' or 'controls'. Out of the 51 patients and 21 healthy blood donors in our cohort, we randomly assigned 51 subjects to represent 'cases' and 21 subjects to represent 'controls'. The 'control' group was used to calculate the autoantibody cutoff levels. The process was repeated 100 times. The permuted datasets revealed a strong bias of the analytical method, specifically identifying high-level autoantibodies ($Z \geq 5$) among the subjects assigned to represent 'cases' (*Figure 1c*).

We also performed a similar analysis limited to the healthy blood donors only, assigning eleven blood donors to represent 'cases' and the remaining ten to represent 'controls'. Autoantibody signals were detected at high level in the blood donors that were assigned to represent 'cases' while those assigned to represent 'controls' did not show autoantibody signals above $Z \geq 3$ (*Figure 1—figure supplement 4*). To better determine how the analytical bias influenced the overall results, we applied a reversed cutoff to our dataset based on the values obtained for the 51 APS1 patients instead of those for the 21 healthy controls. The healthy controls now turned out to show greater numbers of high-level ($Z \geq 5$) autoantibody signals than the APS1 patients, revealing that the skewing effect of the data analysis was dominating the results (*Figure 1d*).

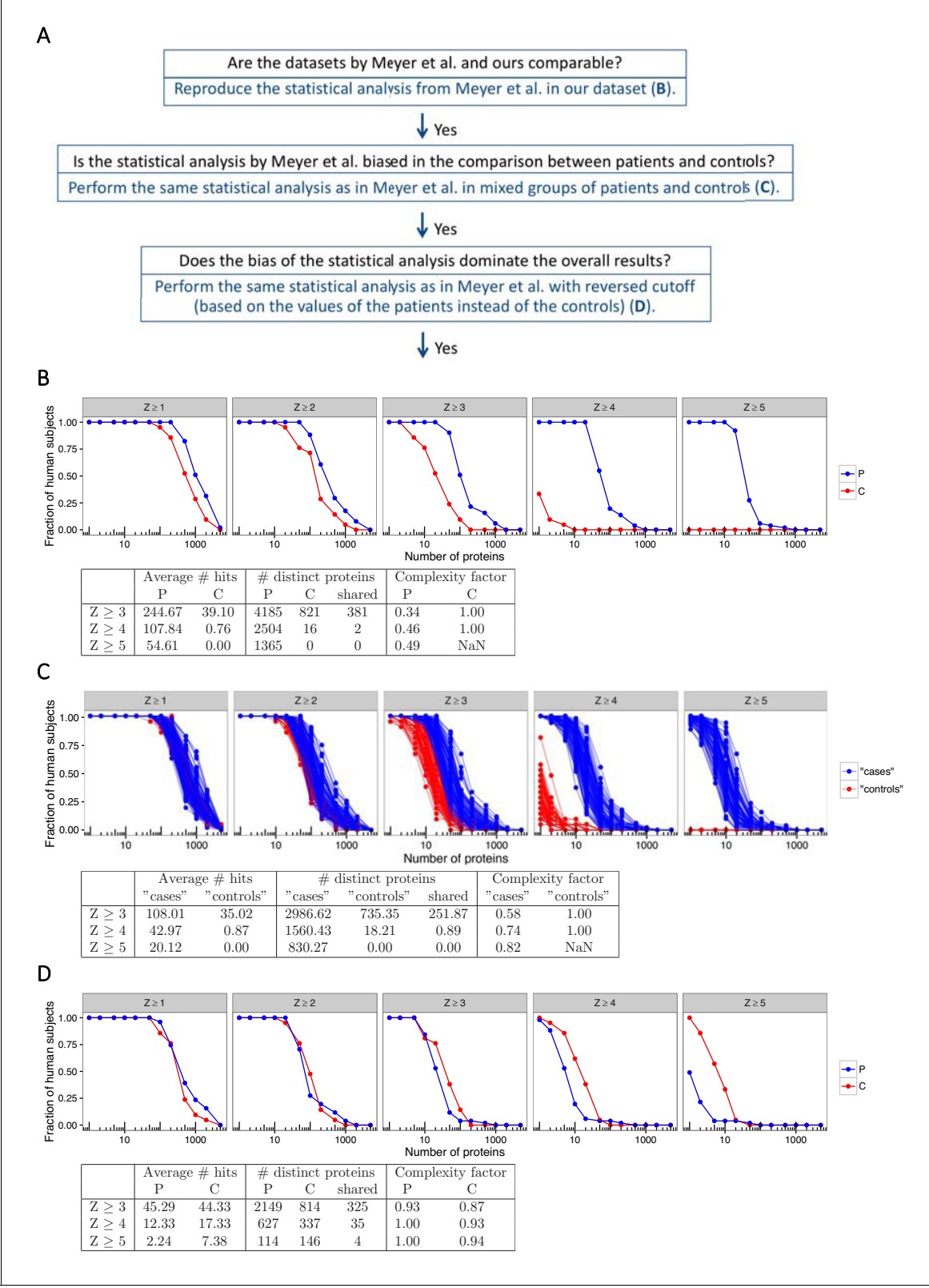

**Figure 1.** The statistical analysis used by Meyer et al. is biased towards overestimation of the number of autoantibody signals in the patient group. (**A**) We used our previously published proteome array dataset for 51 APS1 patients and 21 healthy controls to reevaluate the autoantigen spectrum in APS1. (**B**) To first determine if our dataset and the one by Meyer et al. were comparable, we applied the same statistical analysis as in Meyer et al. to identify autoantibody signals among the APS1 patients and healthy controls. Z-scores were calculated for all individuals over all proteins as the number
*Figure 1 continued on next page*

*Figure 1 continued*

of standard deviations above the mean value, where both mean and standard deviation were calculated based on the 21 healthy controls. Similar to the results by Meyer et al., high-level autoantibody signals ($Z > 5$) were specially observed among the patients. (C) To next determine whether the difference between patients and controls was true our could be explained by a biased statistical analysis we performed the same analysis in groups of subjects randomly assigned to represent 'cases' or 'controls'. Fifty-one subjects were assigned to represent 'cases' and 21 subjects were assigned to represent 'controls', and autoantibody cutoffs were calculated based on the 'controls'. The process was repeated 100 times. High-level autoantibody signals were specifically observed among the subjects assigned to represent 'cases'. Mean values over the 100 permutations are presented in the table. (D) To determine whether the bias of the statistical analysis was dominating the overall results we applied a reversed cutoff, based on the values obtained for the APS1 patients instead of the healthy controls, to the true groups of patients and controls. The controls now showed high-level autoantibodies against greater number of proteins than the patients. In all panels, the average number of hits per individual for patients and healthy controls and the number of distinct proteins targeted in each group are shown for various Z-score cutoffs. The complexity factor was calculated as the number of distinct proteins divided by number of individuals divided by the average number of proteins per individual. NaN – not a number.

DOI: https://doi.org/10.7554/eLife.43578.002

The following figure supplements are available for figure 1:

**Figure supplement 1.** Autoantibody signals for interferons in the protein array.

DOI: https://doi.org/10.7554/eLife.43578.003

**Figure supplement 2.** Simulation where an elevated outlier value at saturation level (65 000 signal units) was introduced in either the APS1 patient group or the control group for the array protein serum albumin.

DOI: https://doi.org/10.7554/eLife.43578.004

**Figure supplement 3.** Simulation where increasingly strong outliers (z-score 2, 3, . . ., 10) were introduced in random normal data for in total 51 'cases' and 21 'controls' respectively, illustrating how the cutoff is affected by elevated outliers in the control group but not by outliers in the case group.

DOI: https://doi.org/10.7554/eLife.43578.005

**Figure supplement 4.** Analyses in permuted data from healthy controls reveal skewing effects in the analysis used by Meyer et al.

DOI: https://doi.org/10.7554/eLife.43578.006

**Figure supplement 5.** Random selection of 10 protein identified as autoantigens in APS1 using the criteria applied by Meyer et al.

DOI: https://doi.org/10.7554/eLife.43578.007

**Figure supplement 6.** Targets identified using the analysis by Meyer et al. are low in signal and do not show enrichment for tissue-specific genes.

DOI: https://doi.org/10.7554/eLife.43578.008

**Figure supplement 7.** No support for widespread autoantigen spectrum in APS1.

DOI: https://doi.org/10.7554/eLife.43578.009

We further found that the targets identified using the data analysis by Meyer et al. were mostly low in signal, and in contrast to the established autoantigen, did not differ from a random sample of the protein panel when looking at the number of tissue-specific genes (*Figure 1—figure supplements 5–6*). When we applied alternative statistical analyses that treated cases and controls in a neutral way to compare the total number of autoantibody signals between the groups, we found only a minor difference between APS1 patients and controls (*Figure 1—figure supplement 7*). The difference between the groups was furthermore mostly explained by the previously identified APS1 autoantigens. Our investigations collectively suggested that the approach to data analysis used by Meyer et al. led to erroneous conclusions regarding the breadth of the antigen spectrum in APS1.

## No association between neutralizing autoantibodies to interferons and type 1 diabetes in APS1

Autoantibodies against type 1 IFNs are present in almost all patients with APS1 and represent a valuable biomarker for the disease, even early in the course of the disease progression (*Meager et al., 2006*). Meyer et al. reported that type I IFN autoantibodies showed neutralizing effect in many but not all patients with APS1. The authors further found an inverse correlation between the capacity to neutralize IFNα and the development of type 1 diabetes in a small number of APS1 patients. Among 21 investigated APS1 patients, sera from all 13 patients without type 1 diabetes neutralized IFNα while sera from all eight patients with type 1 diabetes showed only low or negligible neutralization. No healthy controls or any other non-APS1 controls were included in the experiment. It was proposed that IFN autoantibodies could potentially protect patients with APS1 from developing type 1 diabetes.

We set out to investigate the proposed association between neutralizing type I IFN autoantibodies and type 1 diabetes. We investigated sera from three groups: 16 APS1 patients diagnosed with type 1 diabetes, 14 APS1 patients without type 1 diabetes from Sweden, Finland, the US and the

UK, and five healthy controls. IFN neutralization was studied using a previously described cell-based assay (*Gupta et al., 2016*). Healthy control peripheral blood mononuclear cells were stimulated with IFNα, IFNω or IFNγ in the presence of APS1 patient or healthy control serum. We used the same serum concentration (10%) as previously applied to quantify interferon neutralizing autoantibodies in *Gupta et al. (2016)*. CD14[+] monocytes were then stained for intranuclear phospho-STAT1, which is a marker of IFN-dependent intracellular response. As expected, APS1 sera specifically neutralized IFNα- and IFNω- but not IFNγ-dependent pSTAT1 while healthy control sera did not show neutralization in any of the IFN stimulation assays. As control, serum from a patient with IFNγ autoantibodies and disseminated mycobacterial infection neutralized IFNγ- but not IFNα- or IFNω-dependent pSTAT1, and serum from a patient with IFNα and IFNω autoantibodies and thymoma neutralized IFNα- and IFNω- but not IFNγ-dependent pSTAT1 (data not shown). In contrast to the results by Meyer et al., we found strong neutralization of IFNα- and IFNω-dependent cellular response in all tested APS1 sera, with no difference between patients with type 1 diabetes compared to those without diabetes (*Figure 2*). Our results therefore did not confirm the proposed association between type 1 diabetes and neutralizing type I IFN autoantibodies in APS1.

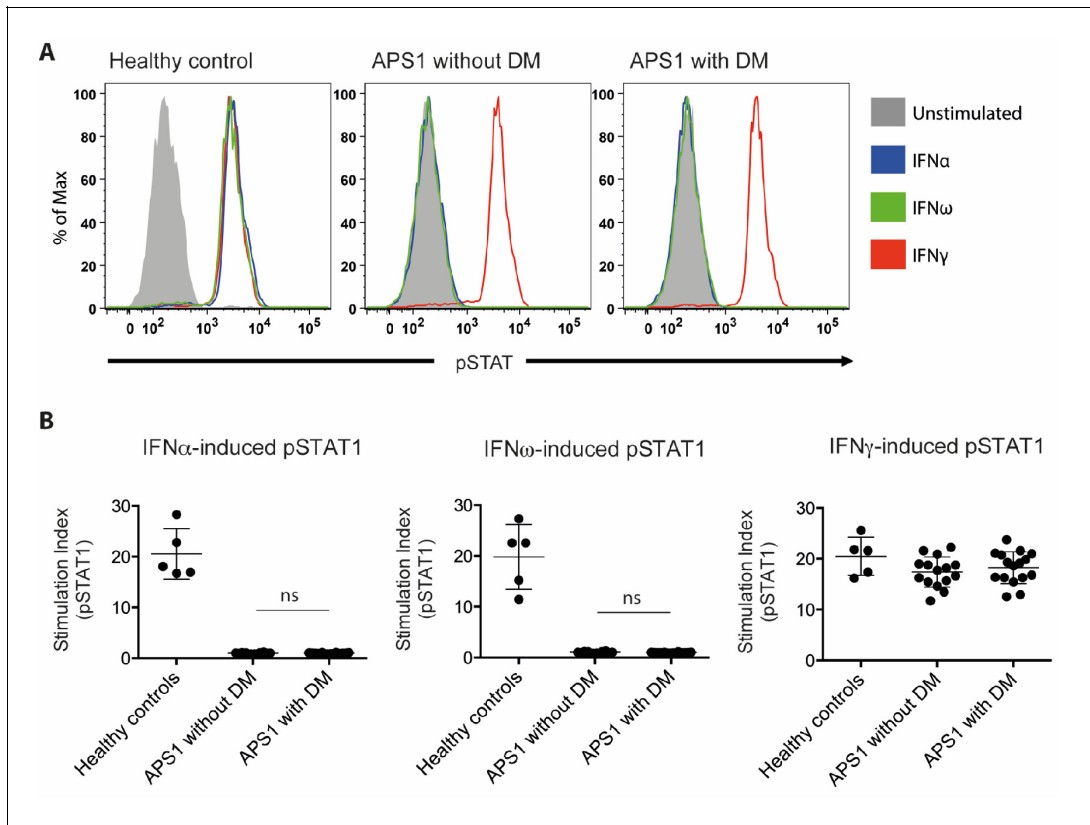

**Figure 2.** No association between neutralizing interferon autoantibodies and type 1 diabetes in APS1. (**A**) Representative FACS plots showing IFNα-induced (blue), IFNω-induced (green), or IFNγ-induced (red) STAT1 phosphorylation in normal PBMC (gating on CD14 +monocytes) in the presence of 10% serum from a healthy control, an APS1 patient without type 1 diabetes (DM), or an APS1 patient with type 1 DM. (**B**) A stimulation index (ratio of stimulated to unstimulated geometric mean fluorescence values for pSTAT1 channel) was calculated for each serum sample tested in response to IFNα, IFNω, and IFNγ. No statistically significant difference was found between the means of APS1 patients with (n = 16) or without type 1 DM (N = 14) for IFNα- or IFNω-induced pSTAT1 (adjusted p value 0.9999 and 0.9976, respectively). IFNγ-induced pSTAT1 was not significantly different between healthy controls, APS1 patients without DM, or APS1 patients with DM. An unmatched 1-way ANOVA was used, correcting for multiple comparisons with Tukey's test. Ns – not significant.

DOI: https://doi.org/10.7554/eLife.43578.010

The following source data is available for figure 2:

**Source data 1.** Neutralizing interferon autoantibodies in APS1 patients with and without type 1 diabetes.
DOI: https://doi.org/10.7554/eLife.43578.011

## Discussion

The study of the autoimmune spectrum in APS1 continues to provide potential novel insights into the general properties of autoimmunity. In this regard, Meyer et al. proposed two new major insights: i) that the spectrum of autoantibodies in APS1 is much broader than previously appreciated; ii) that neutralizing autoantibodies to type 1 IFNs could provide protection against type 1 diabetes in APS1 patients. Using a similar experimental approach on an independent cohort of APS1 subjects we found no evidence to support either conclusion. The finding of a broad autoantigen spectrum in APS1 was based on a biased statistical analysis prone to overestimate the number of autoantibody signals in the patient group. Collectively, we find no evidence to support the previous conclusion that almost half of the proteome is targeted by autoantibodies in APS1 patients and instead find that the autoantibody repertoire is far more restricted.

Studies of APS1 and its mouse model have provided important insights to the molecular mechanisms underlying immunological self-tolerance. The *Aire*-controlled genes have been studied comprehensively in mouse thymic tissue using omics technologies (*Anderson et al., 2002*; *Sansom et al., 2014*; *Venanzi et al., 2008*). Comprehensive studies of autoimmune targets in *AIRE*-deficient subjects represent an equally important source of information for understanding *AIRE*'s role in immune tolerance. Protein array technology provides the opportunity for mapping autoantibody targets on a broad scale. However, an inherent challenge with these types of studies, where thousands of tests are performed simultaneously, is to identify the relevant signals from a background of normal variation and technical noise. Unbiased comparison against controls becomes critical. *Meyer et al. (2016)* used the healthy controls to define normal signal ranges and to identify autoantibody responses. This type of approach is in many situations appropriate. In this case, however, it became a pitfall when a cutoff definition that requires the assumption that the controls show normal signals was used to compare the number of elevated signals between patients and controls. Background signals were unevenly recorded between patients and controls, resulting in the identification of an autoantigen spectrum dominated by the type of stochastic low-level signals that are also present among healthy individuals.

In our previous study of the autoantigen repertoire in APS1 we limited our analyses to autoantibody targets that were shared between multiple patients and thereby could be reliably identified from their statistical association with the patient group (*Landegren et al., 2016b*). Here we used other approaches to also address rare autoantigens in APS1. These studies revealed a only minor difference in the total number of autoantibody signals between APS1 patients and controls, which is in stark contrast with the results by Meyer et al. The healthy control group was small in both our study and the one by Meyer et al. A larger numbers of healthy controls would have provided better means to identify and evaluate the relevance of rare autoantibody signals in APS1 patients.

Meyer et al. also proposed that type I IFN autoantibodies may have disease ameliorating activity and protect patients with APS1 from developing type 1 diabetes, based on the absence of IFN neutralizing autoantibodies in a small number of APS1 subjects with type 1 diabetes (n = 8). In our independent investigation on a larger number of APS1 subjects with type 1 diabetes (n = 16) we could not repeat the reported association. Instead we found strong neutralization of the type I IFN-dependent cellular response in all investigated APS1 sera, with no difference in sera obtained from APS1 patients with or without type 1 diabetes. It is important to consider different explanations why the association between interferon neutralization and type 1 diabetes was not replicated, including differences in methodology. We used an established method and the same serum concentration as previously applied for quantifying neutralizing effect of interferon autoantibodies (*Gupta et al., 2016*). The positive and negative controls in the experiments also verified a reliable detection of neutralizing activity. Our results raise the need for further investigations to establish any association between IFN neutralizing autoantibodies and development of diabetes before embarking on in-depth investigations on targeting type 1 IFNs for the treatment or prevention of type 1 diabetes.

## Materials and methods

**Key resources table**

*Continued on next page*

*Continued*

| Reagent type (species) or resource | Designation | Source or reference | Identifiers | Additional information |
|---|---|---|---|---|
| Reagent type (species) or resource | Designation | Source or reference | Identifiers | Additional information |
| Biological sample (Homo sapiens) | Serum samples from patients with APS1 from Sweden, Norway and Finland, and blood donor controls from Sweden | PMID: 26084804 and 26830021 | | |
| Biological sample (Homo sapiens) | Serum samples from patients with APS1 from USA | PMID: 27588307 | | |
| Biological sample (Homo sapiens) | Blood donor controls from NIH Blood Bank | | | |
| Antibody | Fluorophore-conjugated goat polyclonal anti-human IgG | Thermo Fisher Scientific | Alexa Fluor 647 Goat Anti-Human IgG; Cat#A21445, | Concentration: 1:2000 |
| Antibody | Fluorophore-conjugated mouse monoclonal anti-pSTAT1 | BD Biosciences | Alexa Fluor 647 mouse anti-human Stat1 (pY701); Cat# 612597 | 5 uL/test |
| Antibody | Fluorophore-conjugated mouse monoclonal anti-CD14 | BD Biosciences | FITC mouse anti-human CD14; Cat# 555397 | 2 uL/test |
| Recombinant protein | Recombinant human interferon-alpha 2 | PBL Assay Science | Cat # 11101–2 | Concentration: 10 ng/mL |
| Recombinant protein | Recombinant human interferon-alpha | Peprotech | Cat# 300–02J | Concentration: 10 ng/mL |
| Recombinant protein | Recombinant human interferon-gamma 1b | ActImmune | NDC 42238-111-01 | Concentration: 400 U/mL |
| Commercial assay or kit | Protein microarray | Thermo Fisher Scientific | ProtoArray v5.0 (PAH0525020) | |
| Commercial assay or kit | Blocking buffer kit used in the protein array screening | Thermo Fisher Scientific | Blocking Buffer Kit (PA055) | |
| Software, algorithm | Software used for protein microarray scanning, alignment and data acquisition | | GenePix Pro microarray (v6.1) | |
| Software | FlowJo | | FlowJo (v10.5.3) | |
| Software | Prism | | GraphPad Prism (v6.0) | |

## Study subjects

The patient cohort that was studied using protein arrays included individuals with APS1 from Sweden, Finland and Norway and healthy blood donors from Sweden, as described previously (*Landegren et al., 2016b*; *Landegren et al., 2015*). In the IFN neutralization studies, APS1 patients were also included from a second cohort at NIH. The diagnosis of type 1 diabetes mellitus was based on typical biochemical findings, including glycosuria, elevated plasma glucose levels, decreased plasma levels of insulin and C-peptide, and elevated hemoglobin A1c. Serum autoantibodies against GAD65 were analyzed in all patients. The study was approved by ethics boards of Stockholm (dnr: 2016/2553-31/2) and the NIAID, NIH Clinical Center and NCI Institutional Review Board committees. All patients have provided written informed consent for participation. The study was conducted in accordance with the Helsinki declaration.

## NIH APS1 cohort

Sera were obtained from 12 APS1 patients enrolled in an IRB-approved protocol at the National Institute of Allergy and Infectious Diseases in the United States, as previously described (*Ferre et al., 2016*). Five APS1 patients had type 1 diabetes (representing all patients with type 1 diabetes in the NIH cohort of ~70 APS1 patients), and the remaining 7 APS1 patients did not have type 1 diabetes and were included as randomly-selected controls for functional evaluation of anti-

IFN autoantibodies in serum. The diagnosis of type 1 diabetes mellitus was based on the presence of glycosuria, elevated plasma glucose levels, decreased plasma levels of insulin and C-peptide, and elevated hemoglobin A1c (*Ferre et al., 2016*). Eight patients were female, and five were children under the age of 18; the mean age of the 12 patients was 23.7 years (range 10–66 years). The majority (n = 11) originated from the US and one was from the Isle of Mann in the UK. The most common *AIRE* mutant allele, c.967_979del13, was found in homozygosity in four patients and in heterozygosity in combination with c.769C > T (n = 3), c.522_523ins13 (n = 1), c.1249_1250insC (n = 1), and c.1616C > T (n = 1). One patient was homozygous for c.769C > T and one patient was compound heterozygous for c.769C > T and c.789_789delC (*Ferre et al., 2016*). The most common clinical manifestations in the 12 patients were enamel hypoplasia (91.7%), hypoparathyroidism (83.3%), adrenal insufficiency (83.3%), chronic mucocutaneous candidiasis (75%), hepatitis (66.7%), pneumonitis (58.3%), Sjogren's-like syndrome (41.7%), type 1 diabetes (41.7%), hypogonadism (41.7%), and hypothyroidism (8.3%). Autoantibodies against GAD65 were found in all five patients with type 1 diabetes and in 3 of the seven patients (42.9%) without type 1 diabetes.

## APS1 cohort from Finland and Sweden

Eleven patients with type 1 diabetes and seven patients without type 1 diabetes from Finland and Sweden were included to the study on neutralizing IFN autoantibodies. The diagnosis of type 1 diabetes mellitus was based on typical biochemical findings, including the presence of glycosuria, elevated plasma glucose levels, decreased plasma levels of insulin and C-peptide, and elevated hemoglobin A1c. Seven out of the 18 patients were female. The mean age of the patients was 25 years (range 9–49 years). The most common clinical manifestations in the patients were chronic mucocutaneous candidiasis (100%), adrenal insufficiency (89%), hypoparathyroidism (83%), malabsorption (39%), gonadal insufficiency (33%), pernicious anemia (17%), alopecia (17%), vitiligo (11%) and autoimmune hepatitis (11%). GAD65 autoantibodies were detected in 5 out of 11 (45%) patients with type 1 diabetes and in 2 of the seven patients (29%) without type 1 diabetes.

## Protein array screening

Human protein arrays containing ~9000 full-length human proteins (ProtoArray v5.0, Thermo Fisher) were screened with serum samples from patients with APS1 and healthy controls. The arrays were probed with serum at a concentration of 1:2000, and otherwise followed Invitrogen's protocol for *Immune Response BioMarker Profiling*. A GenePix 4000B microarray scanner and the GenePix Pro microarray (v6.1) software was used for scanning, alignment and data acquisition. The proteome array dataset has been described previously (*Landegren et al., 2016b*; *Landegren et al., 2015*). The protein array screening experiment was not repeated. Suspected printing contamination artifacts were identified and excluded as previously described (*Landegren et al., 2016b*). No outliers were excluded.

## Data analysis

Fluorescence intensities were background corrected and the mean value was computed over duplicate spots. Print-order artifacts were removed as follows: all spots with a Spearman correlation >0.8 to a spot with higher intensity that was printed right before were excluded. Values were log10-transformed and robust linear modeling (RLM) was applied to normalize the data based on human IgG and anti-human IgG controls as described previously (*Sboner et al., 2009*). RLM was implemented in the rlm function in the R package MASS. When more than one spot on the array was associated with the same gene name the mean value over the spots was calculated, such that each gene name was only represented once. The Z-scores were computed as the number of standard deviations away from the mean, where both mean and standard deviation were computed based on a set of 'controls'. The Z-scores were computed for each protein separately.

We also calculated a robust Z-score based on trimmed mean and standard deviations, where the most extreme 10% of the values at each end were excluded when computing the mean and standard deviation. Enrichment of targets in any of the *Human Protein Atlas* classifications was measured using a hypergeometric test. The expression annotation categories were defined according to the following: *tissue enriched* – at least five-fold higher mRNA levels in a particular tissue as compared to all other tissues), *group enriched* - at least five-fold higher mRNA levels in a group of 2–7 tissues

and *tissue enhanced* - at least five-fold higher mRNA levels in a particular tissue as compared to average levels in all tissues. The following proteins in the ProtoArray panel were treated as known autoantigens in the analyses: CYP1A2, DDC, GAD1, GAD2, GIF, IFNA1, IFNA13, IFNA14, IFNA16, IFNA17, IFNA2, IFNA21, IFNA4, IFNA5, IFNA6, IFNA8, IFNW1, IL17A, IL22, KCNRG, MAGEB2, PDILT, TGM4, TPH1 and TSGA10.

### Functional evaluation of interferon autoantibodies

Healthy control peripheral blood mononuclear cells (PBMC; $1 \times 10^7$/mL) were either left unstimulated or stimulated for 15 min at 37°C with IFNα (10 ng/mL), IFNω (10 ng/mL), or IFNγ (400 U/mL) in the presence of 10% patient or control serum, as previously described (*Gupta et al., 2016*). Monocytes were identified by CD14 surface staining (BD Biosciences) before being fixed and permeabilized for intranuclear phospho-STAT1 (Y701; BD Biosciences). Data were collected using an LSRFortessa (BD Biosciences), analyzed using FlowJo software, and graphed with Prism6 software (GraphPad).

## Acknowledgements

We thank Cindy Wong for critical review of the manuscript. This work was supported in part by the Division of Intramural Research of the NIAID/NIH

## Additional information

### Competing interests

Michael Snyder: Serves as founder and consultant for Personalis, is a member of the scientific advisory board of GenapSys, and a consultant for Illumina. Olle Kämpe: Is a board member of Olink Bioscience. The other authors declare that no competing interests exist.

### Funding

| Funder | Author |
| --- | --- |
| Swedish Research Council Formas | Olle Kämpe |
| Novo Nordisk Foundation | Olle Kämpe |
| ALF | Olle Kämpe |
| Leona M. and Harry B. Helmsley Charitable Trust | Mark Anderson |
| Division of Intramural Research, National Institute of Allergy and Infectious Diseases | Michail Lionakis |
| Swedish Research Council | Olle Kämpe |
| Swedish Society for Medical Research | Nils Landegren |
| The Swedish Association of Endocrinologists | Nils Landegren |
| Swedish Research Council | Nils Landegren |
| Knut and Alice Wallenberg Foundation | Olle Kämpe |

The funders had no role in study design, data collection and interpretation, or the decision to submit the work for publication.

### Author contributions

Nils Landegren, Conceptualization, Formal analysis, Writing—original draft; Lindsey B Rosen, Formal analysis, Investigation, Writing—review and editing, Analysis of interferon neutralising

autoantibodies; Eva Freyhult, Formal analysis, Writing—review and editing, Bioinformatic analysis of protein array data; Daniel Eriksson, Formal analysis, Writing—review and editing, Bioinformatic analysis; Tove Fall, Gustav Smith, Petter Brodin, Writing—review and editing, Critical review of the manuscript; Elise M N Ferre, Project administration, Writing—review and editing, Clinical characterisation of APS1 patients; Donald Sharon, Investigation, Writing—review and editing; Michael Snyder, Resources, Funding acquisition, Writing—review and editing; Michail Lionakis, Mark Anderson, Olle Kämpe, Conceptualization, Supervision, Funding acquisition, Writing—review and editing

## Author ORCIDs
Nils Landegren (iD) http://orcid.org/0000-0002-6163-9540
Daniel Eriksson (iD) http://orcid.org/0000-0001-5473-3312
Elise M N Ferre (iD) http://orcid.org/0000-0002-3285-7768
Petter Brodin (iD) http://orcid.org/0000-0002-8103-0046
Olle Kämpe (iD) http://orcid.org/0000-0001-6091-9914

## Ethics
Human subjects: The study was approved by ethics boards of Stockholm (dnr: 2016/2553-31/2) and the NIAID, NIH Clinical Center and NCI Institutional Review Board committees. All patients have provided written informed consent for participation. The study was conducted in accordance with the Helsinki declaration.

## Decision letter and Author response
Decision letter https://doi.org/10.7554/eLife.43578.017
Author response https://doi.org/10.7554/eLife.43578.018

# Additional files

## Supplementary files
• Transparent reporting form
DOI: https://doi.org/10.7554/eLife.43578.012

## Data availability
Source data has been provided Figure 2. Protein array data is available at ArrayExpress with accession E-MTAB-8097.

The following dataset was generated:

| Author(s) | Year | Dataset title | Dataset URL | Database and Identifier |
|---|---|---|---|---|
| Nils Landegren, Donald Sharon, Michael Snyder, Olle Kämpe | 2019 | Protein microarray-based autoantibody screening in 51 APS1 patients and 21 healthy controls | https://www.ebi.ac.uk/arrayexpress/experiments/E-MTAB-8097/ | ArrayExpress, E-MTAB-8097 |

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
