## [Decision Letter]

Thank you for submitting your article "Comment on 'AIRE-deficient patients harbor unique high-affinity disease-ameliorating autoantibodies'" to *eLife* for consideration as Scientific Correspondence. Your article, and the response to it from Kisand, Hayday and colleagues, have been reviewed by two peer reviewers (who have opted to remain anonymous), and the evaluation has been overseen by a Deputy Editor (Detlef Weigel) and the *eLife* Features Editor (Peter Rodgers).

I am pleased to be able to tell you that we have decided to accept for publication both your article and the response to it. The substantive comments in this decision letter will also be published as part of your article (and likewise for the response): you will be able to check these comments when you check the proof of your article.

You also need to make the following changes to your article:

[...]

Note: The editor handling this manuscript asked the reviewers to answer a number of questions: these questions are shown below in italics; the comments from the reviewers are shown in Roman.

Reviewer #1:

1) Are the elements of the present submission significant enough to merit publication in eLife? Do you have any substantive concerns about the elements of the article that challenge the findings of the original publication? Are the other elements of challenge significant enough to merit publication in eLife? Do you have any substantive concerns about these other elements?

There are two points of contention raised by the Landegren article in relation to the Meyer et al., 2016 Cell paper. The first is whether the extent of autoantigens in APECED patients is truly as broad as claimed by the Meyer publication. This is well challenged by the article, raising a 'data interpretation' debate. Landegren goes to great lengths to prove how the same data can be analyzed to show a broad range of serum autoreactivity or a restricted set of autoantigens. In an age of big data, this is entertaining and enlightening. Their concern is 100% warranted. The problem that is raised is that if one uses a small set of controls that are relatively homogeneous to define Z score thresholds for thousands of analytes, the analysis will favor outliers in a larger patient group. This is obvious, but not well understood. One simply should not use a small number of controls to define thresholds in these sorts of assays. Or if one does, then there must be a validation with a larger set or different assays. The point is well demonstrated. Landegren could have taken this further and more generally by making the point that one should not use a small number of controls to define thresholds as they simply are NOT sufficiently representative of the range of signals in health - that would be helpful for readers. What is perhaps less well done is to allow room for the possibility that there may be a broad reactivity since neither the original study nor the challenging analysis actually has sufficient controls to appropriately do the Z score approach and the alternative Landegren Fishers test approach is conservative. In other words, neither can actually conclude their view on broad vs restricted. Landegren should be critical of their own study by stating that they also had few controls and, therefore, used a more conservative approach to define the threshold, but that this is likely to be biased in the opposite direction to that of Meyer.

The second point of contention is the relationship of type 1 IFN neutralizing autoantibodies to diabetes in the APECED patients. Here, the challenge may be warranted but less convincing. It is important to examine a second cohort since the numbers in the Meyer paper were small. However, as pointed out in the formal response, Landegren failed to sufficiently quantify neutralization. They can conclude that the APECED patients who develop diabetes in their cohort have type 1 IFN neutralizing autoantibodies, but they cannot comment on whether the titer is the same or different to the APECED patients without diabetes since they have reached assay saturation in both groups. This is a pity since a suitably performed validation experiment would be a valuable contribution.

Reviewer #2:

1) Are the elements of the present submission significant enough to merit publication in eLife? Do you have any substantive concerns about the elements of the article that challenge the findings of the original publication? Are the other elements of challenge significant enough to merit publication in eLife? Do you have any substantive concerns about these other elements?

i) Are the elements of the present submission significant enough to merit publication in eLife?

Yes. The two main novel findings in the Cell paper, which have generated wide interest, appear not to be correct. The third finding that anti-interferon antibodies, which were already known to be very prevalent in AIRE deficiency, are of very high affinity due to affinity maturation, is not challenged by Landegren.

ii) Do you have any substantive concerns about the elements of the article that challenges the findings of the original publication?

No, the analysis performed by Landegren provide compelling evidence that the original publication in Cell employed a flawed normalization of the analysis of antibody binding to protein microarrays. As a result, the main conclusion in the Cell paper about the much wider breakdown of tolerance in AIRE deficiency, with antibodies recognizing private specificities unique to one or two patients, is erroneous.

In rebuttal, the authors of the Cell paper claim that a sampling of the thousands of private specificities were validated. There was no evidence for this in the published study, and unconvincing preliminary evidence here.

iii) Are the other elements of challenge significant enough to merit publication in eLife?

Yes. The second main novel conclusion of the Cell paper was that neutralizing antibodies to interferon were absent from the sera of AIRE deficient patients with Type 1 diabetes compared to those without. Here, studying a larger cohort, Landegren et al find neutralizing antibodies in AIRE deficient patients with or without Type 1 Diabetes. It is possible this reflects differences in assay methodology, but both methods appear equally valid and unlikely to explain the different results.

In rebuttal, the original authors point out that the data in the Cell paper were from serum titrations, and that if they repeat the experiments with serum diluted 1/10 they also now find neutralizing antibodies in both groups. In the Cell paper, the mean IC50 of neutralizing antibodies in patients without type 1 diabetes was on the order of 1/100,000, at which point there was a difference in titre. However in the data in the rebuttal, the difference in inhibitory activity is only apparent at one dilution (1/50) and not at 1/250 or 1/10. Since the rebuttal now also shows neutralizing antibodies are present in Type 1 diabetes patients at relatively small differences in titre, and since serum is undiluted in vivo, this appears to warrant a revision to the original conclusions in Cell that "those with T1D showed only low or negligible neutralization."

It is nevertheless difficult to extrapolate serum neutralization from in vitro to in vivo, so it would be important for Landegren et al to test interferon alpha neutralization at different serum dilutions to determine if there is a difference in titre in their cohort.

iv) Do you have any substantive concerns about these other elements?

No.

---

## [Author Response]

We repeat the reviewers’ points here in italic, and include our replies point by point, as well as a description of the changes made, in Roman.

Reviewer #1:

1) Are the elements of the present submission significant enough to merit publication in eLife? Do you have any substantive concerns about the elements of the article that challenge the findings of the original publication? Are the other elements of challenge significant enough to merit publication in eLife? Do you have any substantive concerns about these other elements?There are two points of contention raised by the Landegren article in relation to the Meyer 2016 Cell paper. The first is whether the extent of autoantigens in APECED patients is truly as broad as claimed by the Meyer publication. This is well challenged by the article, raising a 'data interpretation' debate. Landegren goes to great lengths to prove how the same data can be analyzed to show a broad range of serum autoreactivity or a restricted set of autoantigens. In an age of big data, this is entertaining and enlightening. Their concern is 100% warranted. The problem that is raised is that if one uses a small set of controls that are relatively homogeneous to define Z score thresholds for thousands of analytes, the analysis will favor outliers in a larger patient group. This is obvious, but not well understood. One simply should not use a small number of controls to define thresholds in these sorts of assays. Or if one does, then there must be a validation with a larger set or different assays. The point is well demonstrated. Landegren could have taken this further and more generally by making the point that one should not use a small number of controls to define thresholds as they simply are NOT sufficiently representative of the range of signals in health - that would be helpful for readers. What is perhaps less well done is to allow room for the possibility that there may be a broad reactivity since neither the original study nor the challenging analysis actually has sufficient controls to appropriately do the Z score approach and the alternative Landegren Fishers test approach is conservative. In other words, neither can actually conclude their view on broad vs restricted. Landegren should be critical of their own study by stating that they also had few controls and, therefore, used a more conservative approach to define the threshold, but that this is likely to be biased in the opposite direction to that of Meyer.

REPLY: We appreciate the Reviewer’s recognition of efforts to bring clarity to the conflicting reports regarding the autoantigen repertoire in *AIRE*-deficient patients. It is encouraging to read that the Reviewer finds our points well demonstrated and enlightening.

We agree with the Reviewer that the small number of controls is an important limitation of both the study by Meyer et al. and ours. A larger number of controls would indeed have provided better means to evaluate what is truly patient-specific and what can be explained by normal variation. This would be especially important for determining the relevance (if any) of the type of rare autoantibody signals that are observed both among patients and healthy controls.

Despite this limitation, we are able to conclude the following:

1) Only a very restricted set of proteins (<1% of the protein panel) are targeted by autoantibodies in multiple APS1 patients. These major autoantigens, which can be readily identified by their statistical association with the patient group, are described in our previous article (Landegren et al., 2016).

2) There is no support for the Cell paper claim of broad antigen spectrum in APS1. Our analyses of permutated data clearly shows that their observed difference in numbers of autoantibody signals between patients and controls can be explained by a bias in the data analysis.

3) The total number autoantibody signals (also including rare autoantigens) is only slightly higher in APS1 compared to healthy control, which is in stark contrast with the Cell paper claim. When we apply different approaches that treat patients and controls in a neutral way to count the total number of autoantibody signals (either using Z-scores based on the values for both patients and controls where the most extreme values at each end have been excluded, or simply using fixed signal intensity cutoffs) we only find small difference between the patient group and the control group. This difference was furthermore mostly explained by the previously described autoantigens (Figure 1-figure supplement 7).

Changes made to the manuscript:

We have now added the following to the Discussion, explaining limitations of our study of the autoantigen repertoire:

In our previous study of the autoantigen repertoire in APS1 we limited our analyses to autoantibody targets that were shared between multiple patients and thereby could be reliably identified from their statistical association with the patient group (Landegren et al., 2016b). Here we used other approaches to also address rare autoantigens in APS1. These studies revealed a only minor difference in the total number of autoantibody signals between APS1 patients and controls, which is in stark contrast with the results by Meyer et al. The healthy control group was small in both our study and the one by Meyer et al. A larger numbers of healthy controls would have provided better means to identify and evaluate the relevance of rare autoantibody signals in APS1 patients.

The second point of contention is the relationship of type 1 IFN neutralizing autoantibodies to diabetes in the APECED patients. Here, the challenge may be warranted but less convincing. It is important to examine a second cohort since the numbers in the Meyer paper were small. However, as pointed out in the formal response, Landegren failed to sufficiently quantify neutralization. They can conclude that the APECED patients who develop diabetes in their cohort have type 1 IFN neutralizing autoantibodies, but they cannot comment on whether the titer is the same or different to the APECED patients without diabetes since they have reached assay saturation in both groups. This is a pity since a suitably performed validation experiment would be a valuable contribution.

REPLY: We here sought to provide an independent validation of the strong claims in the Cell paper regarding the existence of disease ameliorating autoantibodies in APS1. Specifically, we sought to repeat the observation that “patients without T1D collectively neutralized all IFNa subtypes, whereas those with T1D showed only low or negligible neutralization”.

In our validation study, we applied an established method to investigate autoantibody-mediated interferon neutralization in a larger cohort of APS1 patients with and without T1 diabetes. We used the same serum concentration as applied in the article by Gupta et al. for quantifying neutralization effect of interferon autoantibodies. The performance of the assay was verified using positive control sera (from a patient with IFNγ autoantibodies and disseminated mycobacterial infection, and from a patient with IFNα and IFNω autoantibodies and thymoma). In contrast to the Cell paper we included healthy blood donor sera as negative controls in our experiment, which would allow us to determine whether or not the APS1 without T1 diabetes truly showed negligible neutralization as previously claimed.

Our validation experiment showed strong and specific neutralization of IFNα- and IFNω in all tested APS1 sera, with no difference between patients with type 1 diabetes compared to those without diabetes. The results thus did not support the reported association with T1 diabetes status. Furthermore, our results from comparing APS1 to healthy control sera were in direct conflict with the Cell paper claim of low or negligible interferon neutralization in APS1 patients with T1 diabetes.

We understand from reading the comments of Reviewer 2 that the authors of the Cell paper can now only detect a difference in neutralization between patients with and without diabetes at certain serum dilutions. Hopefully further independent replication studies will be performed in light of the results presented in our Comment and in the Response to our Comment.

Changes made to the manuscript:

We have added the following to the Results, to clarify our reasoning behind the design of our experiments:

“We used the same serum concentration (10%) as previously applied to quantify interferon neutralizing autoantibodies in the study by Gupta et al.”

We have also added the following to the Discussion regarding limitations of our investigation:

“It is important to consider different explanations why the association between interferon neutralization and type 1 diabetes was not replicated, including differences in methodology. We used an established method and the same serum concentration as previously applied for quantifying neutralizing effect of interferon autoantibodies (Gupta et al., 2016). The positive and negative controls in the experiments also verified a reliable detection of neutralizing activity.”

Reviewer #2:

1) Are the elements of the present submission significant enough to merit publication in eLife? Do you have any substantive concerns about the elements of the article that challenge the findings of the original publication? Are the other elements of challenge significant enough to merit publication in eLife? Do you have any substantive concerns about these other elements?i) Are the elements of the present submission significant enough to merit publication in eLife?Yes. The two main novel findings in the Cell paper, which have generated wide interest, appear not to be correct. The third finding that anti-interferon antibodies, which were already known to be very prevalent in AIRE deficiency, are of very high affinity due to affinity maturation, is not challenged by Landegren.ii) Do you have any substantive concerns about the elements of the article that challenges the findings of the original publication?No, the analysis performed by Landegren provide compelling evidence that the original publication in Cell employed a flawed normalization of the analysis of antibody binding to protein microarrays. As a result, the main conclusion in the Cell paper about the much wider breakdown of tolerance in AIRE deficiency, with antibodies recognizing private specificities unique to one or two patients, is erroneous. In rebuttal, the authors of the Cell paper claim that a sampling of the thousands of private specificities were validated. There was no evidence for this in the published study, and unconvincing preliminary evidence here.

REPLY: It is encouraging to read the strong recognition of our work. We share the Reviewer’s view on the autoantibody results presented in the Cell paper.

iii) Are the other elements of challenge significant enough to merit publication in eLife?Yes. The second main novel conclusion of the Cell paper was that neutralizing antibodies to interferon were absent from the sera of AIRE deficient patients with Type 1 diabetes compared to those without. Here, studying a larger cohort, Landegren et al find neutralizing antibodies in AIRE deficient patients with or without Type 1 Diabetes. It is possible this reflects differences in assay methodology, but both methods appear equally valid and unlikely to explain the different results.In rebuttal, the original authors point out that the data in the Cell paper were from serum titrations, and that if they repeat the experiments with serum diluted 1/10 they also now find neutralizing antibodies in both groups. In the Cell paper, the mean IC50 of neutralizing antibodies in patients without type 1 diabetes was on the order of 1/100,000, at which point there was a difference in titre. However in the data in the rebuttal, the difference in inhibitory activity is only apparent at one dilution (1/50) and not at 1/250 or 1/10. Since the rebuttal now also shows neutralizing antibodies are present in Type 1 diabetes patients at relatively small differences in titre, and since serum is undiluted in vivo, this appears to warrant a revision to the original conclusions in Cell that "those with T1D showed only low or negligible neutralization."It is nevertheless difficult to extrapolate serum neutralization from in vitro to in vivo, so it would be important for Landegren et al to test interferon alpha neutralization at different serum dilutions to determine if there is a difference in titre in their cohort.iv) Do you have any substantive concerns about these other elements?No.

REPLY: We thank the Reviewer for the valuable comments.

In our validation experiments we used an established method and the same serum concentration as previously applied for measuring neutralizing activity of interferon autoantibodies. The performance of the assay was verified using positive and negative controls (which was lacking in the original study). Our investigations revealed strong and specific neutralization of IFNα- and IFNω in all tested APS1 sera, with no difference between patients with type 1 diabetes compared to those without diabetes. The results thus did not support the reported association with T1 diabetes status.

It is surprising that difference in neutralization would only be detected in a very narrow range of concentration. We hope that further independent replication studies will be performed in light of the results presented in our Comment and in the Response to our Comment.

The following has been added to the Results to explain our reasoning behind choice of experimental setup:

“We used the same serum concentration (10%) as previously applied to quantify interferon neutralizing autoantibodies in the study by Gupta et al.”

We have also added the following to the Discussion to emphasize limitations:

“It is important to consider different explanations why the association between interferon neutralization and type 1 diabetes was not replicated, including differences in methodology. We used an established method and the same serum concentration as previously applied for quantifying neutralizing effect of interferon autoantibodies (Gupta et al., 2016). The positive and negative controls in the experiments also verified a reliable detection of neutralizing activity.”

References:

Gupta, S., Tatouli, I.P., Rosen, L.B., Hasni, S., Alevizos, I., Manna, Z.G., Rivera, J., Jiang, C., Siegel, R.M., Holland, S.M.*, et al.* (2016). Distinct Functions of Autoantibodies Against Interferon in Systemic Lupus Erythematosus: A Comprehensive Analysis of Anticytokine Autoantibodies in Common Rheumatic Diseases. Arthritis & Rheumatology *68*, 1677-1687.

Landegren, N., Sharon, D., Freyhult, E., Hallgren, A., Eriksson, D., Edqvist, P.H., Bensing, S., Wahlberg, J., Nelson, L.M., Gustafsson, J.*, et al.* (2016). Proteome-wide survey of the autoimmune target repertoire in autoimmune polyendocrine syndrome type 1. Scientific Reports *6*, 20104.